# The Interaction of Glycemia with Anxiety and Depression Is Related to Altered Cerebellar and Cerebral Functional Correlations

**DOI:** 10.3390/brainsci13071086

**Published:** 2023-07-18

**Authors:** Grace E. Shearrer

**Affiliations:** Department of Family and Consumer Sciences, Neuroscience Program, School of Computing, College of Agriculture, Life Sciences and Natural Resources, University of Wyoming, Laramie, WY 82070, USA; gshearre@uwyo.edu

**Keywords:** depression, anxiety, glycemia, fMRI

## Abstract

Depression, type 2 diabetes (T2D), and obesity are comorbid, and prevention and treatment of all three diseases are needed. We hypothesized an inverse relationship between the connectivity of the cingulo-opercular task control network with the somatosensory mouth network and the interaction between HbA1c and depression. Three-hundred and twenty-five participants (BMI: 26.11 ± 0.29; Achenbach adult self-report (ASR) DSM depressive problems T-score (depression): 54.60 ± 6.77; Age: 28.26 ± 3.90 y; adult self-report anxiety and depression scale (anxiety and depression): 54.69 ± 7.27; HbA1c: 5.26 ± 0.29; 68% white) were sampled from the Human Connectome Project 1200 subjects PTN release. Inclusion criteria were: four (15 min) resting state fMRI scans; BMI; hemoglobin A1c (HbA1c); and complete adult self-report data. The following models were run to assess the connectivity between 15 independent fMRI components: the interaction of depression with HbA1c; anxiety and depression with HbA1c; depression with BMI; and anxiety and depression with BMI. All models were corrected for a reported number of depressive symptoms, head motion in the scanner, age, and race. Functional connectivity was modeled in FSLNets. Corrected significance was set at pFWE < 0.05. The interaction HbA1c and anxiety and depression was positively related to the connectivity of the cerebellum with the visual network (t = 3.76, pFWE = 0.008), frontoparietal network (t = 3.45, pFWE = 0.02), and somatosensory mouth network (t = 4.29, pFWE = 0.0004). Although our hypotheses were not supported, similar increases in cerebellar connectivity are seen in patients with T2D and overall suggest that the increased cerebellar connectivity may be compensatory for an increasingly poor glycemic control.

## 1. Introduction

Increasingly, obesity and metabolic dysfunction are recognized as risk factors not only for physical health problems (cardiovascular disease, certain cancers, stroke), but also for mental health problems [1]. The comorbidity between excess weight and depression appears to be bidirectional. People with obesity are at a 55% increased risk of developing depression, and people with depression are at a 58% increased risk of becoming obese [2]. Further, depression is strongly comorbid with anxiety, with nearly 46% of people with depression also having one or more anxiety disorder [3]. Unsurprisingly, anxiety is also related to metabolic syndrome, a constellation of closely related cardiovascular risk factors, including hyperglycemia [4]. Usually discussed in the context of diabetes, hyperglycemia is defined as fasting blood glucose concentrations greater than 125 mg/dL and is a result of poor glycemic control (optimal serum glucose concentration) [5]. Adequate insulin production, insulin sensitivity, glucose intake, along with a variety of other endocrine factors (glucagon like peptide-1, glucagon) establish glycemic control and are related to diabetes severity [5]. Depression and anxiety have been examined across multiple diabetes subgroups including individuals with type 1 [6], type 2 [7], and gestational diabetes [8]. Regardless of diabetes type, anxiety and depression are associated with poorer glycemic control [6,7,8]. Overall, research indicates that mental and physical health are closely linked, particularly between hyperglycemia and depression and anxiety. However, since most research is on individuals with a clinical pathology (type 2 diabetes (T2D); type 1 diabetes (T1D)), it is difficult to separate anxiety and depression’s association with the disease state or a feeling of loss, hopelessness, and/or anxiety due to the diagnosis [9].

Despite the strong correlations between depression, anxiety, obesity, and glycemia, few studies have examined the interaction of these comorbidities in the brain. Animal studies have shown that a prolonged high-fat diet intake increased depression and anxiety behaviors and altered reward and feeding region activity in the brain [10,11]. In humans, youth with insulin resistance show increased depressive symptoms, and both insulin resistance and depression were related to a higher functional connectivity between the anterior cingulate cortex (a region implicated in motivation, decision making, learning, and error monitoring [12]) and the hippocampus (a region important for memory and learning) [13]. In youth with depression and who experienced childhood abuse exhibited a higher fasting glucose [14]. Further, fasting glucose was related to an increased connectivity between the amygdala and precuneus [14]. Meta-analyses show an increased insula activity associated with anxiety [15], a decreased insula-to-medial frontal gyrus and supplemental motor areas connectivity associated with T2D (using a seed ROI in the insula) [16], and a decreased insula response in individuals with obesity [17]. On the other hand, the caudate, a region implicated in reward-related behaviors, shows a decreased activity after the receipt of a reward in individuals with depression [18] and obesity [19]. Together, this suggests common brain regions with an altered activity across metabolic (T2D and obesity) and mental (anxiety and depression) pathologies.

The present analysis aimed to evaluate the interactions between hemoglobin A1c (HbA1c, a metric of glycemic control), body mass index (BMI), anxiety and depression together, and depression in a large cohort of healthy adults from the Human Connectome Project (HCP) [20]. Assessing the interactions between metabolic (HbA1c, BMI) and mental (anxiety and depression) health in a healthy sample circumvents potential confounding due to diagnosis. Further, this allows us to examine pre-pathological associations to better inform preventative efforts. We used independent component analysis (ICA) to isolate brain regions of interest. We then evaluated the association between the ICA maps and the interaction of Achenbach adult self-report (ASR) depression and anxiety percentiles and ASR depression problems T-scores with HbA1c and BMI, respectively. Analyzing both HbA1c and BMI allows us to delineate glycemia-related changes from BMI-related changes which are often confounded in T2D analyses. Similarly, we chose to analyze both depression and anxiety and depression alone to evaluate similarities and differences between the two. We have previously published using similar methods within the HCP data to evaluate differences in connectivity based on HbA1c [21], menstrual onset [22], and BMI [23]. Based on meta-analytical data, we hypothesized that depression and anxiety would interact with HbA1c in the insula [15,16,17]. Specifically, that higher depression and anxiety and high HbA1c would be related to a lower connectivity between the insula and anterior cingulate. Further, we hypothesized that depression would interact with BMI, resulting in higher depression and BMI associated with lower connectivity between the caudate and a cingulo-opercular network [18,19].

## 2. Materials and Methods

### 2.1. Sample

Participants were selected from the HCP 1200 parcellation, timeseries, and netmats (PTN) release [20]. All participants in the HCP presented written informed consent approved by both the Washington University in St. Louis and the University of Minnesota Institutional Review Boards. For the present analysis, participants were included if they had complete resting-state functional magnetic resonance imaging (rs-fMRI) data, BMI data, HbA1c, Achenbach adult self-report (ASR) anxiety and depression percentile score adjusted for gender and age (AnxDepr), ASR DSM depressive problems (scale) gender- and age-adjusted T-score (Depr), and semi-structured assessment for the genetics of alcoholism (SSAGA) depressive symptoms data. Of the 1206 total participants, 325 participants met the inclusion criteria. Sample characteristics are summarized in Table 1.

### 2.2. Data Description and Preprocessing

Data collection, preprocessing, and the development of the PTN release is well documented [24,25,26]. Of note, participants completed 4 rs-fMRI scans over two days on a 3T Siemens Skyra magnet, totaling 58 min and 12 s of rs-fMRI data per participant. Participants were asked to keep their eyes open, relaxed, and viewed a light crosshair on a dark background projected into their field of view. The PTN release was extensively preprocessed, and no additional preprocessing was performed locally [24].

Non-imaging, behavioral and physical measures are detailed here [27]. The HCP used the Achenbach adult self-report (ASR) to measure anxiety and depression percentiles as well as depression percentile scores adjusted for age and gender. The ASR is a validated instrument to assess adult psychopathology and is used in clinical and research settings [28]. The semi-structured assessment for the genetics of alcoholism (SSAGA-II) was used to determine the number of depressive symptoms. The SSAGA-II assesses the physical, psychological, and social aspects of substance use as well as other psychiatric disorders, including depression [29]. Height and weight were self-reported. Details related to the blood draw and HbA1c measurements can be found here [30].

### 2.3. Group ICA

Dense connectomes were generated using incremental principal components analysis and were parcellated using group-independent independent components analysis (ICA) to create 15 spatial ICA network maps [31]. The 15 ICA parcellation was chosen to minimize multiple comparisons and generate network maps similar to Yeo’s 17 network cortical parcellation [32]. Because multiple components in a network may include an overlap of anatomical regions with other components, independent components (ICs) will be referred to by their number and anatomical region(s) (e.g., IC 3 primary visual network).

### 2.4. Individual Component Timeseries and Creation of Netmats

Participant imaging timeseries were concatenated and spatially mapped to the corresponding network map. Dual regression was used to regress each group network map against individual timeseries to create participant network matrices (netmats). For consistency of this analysis with previous work, IC 9 was ignored as noise [21].

### 2.5. Statistical Analyses

Imaging analyses were performed in FSLNets (Version 0.6, FMRIB, Oxford, UK). ICs were correlated with each other to produce a 15 × 15 individual correlation matrix with normalized covariances. Four general linear models were fitted to the functional correlations (FCs) between ICs:*Model* 1. *functional correlations* = [*HbA1c* × *AnxDepr*] + *AnxDepr* + *HbA1c* + *BMI* + *DeprSymp* + *head* motion + *Age* (*y*) + *race*
*Model* 2. *functional correlations* = [*BMI* × *AnxDepr*] + *AnxDepr* + *HbA1c* + *BMI* + *DeprSymp* + *head motion* + *Age* (*y*) + *race*
*Model* 3. *functional correlations* = [*HbA1c* × *Depr*] + *Depr* + *HbA1c* + *BMI* + *DeprSymp* + *head motion* + *Age* (*y*) + *race*
*Model* 4. *functional correlations* = [*BMI* × *Depr*] + *Depr* + *HbA1c* + *BMI* + *DeprSymp* + *head motion* + *Age* (*y*) + *race*

To correct for potential false positives, non-parametric permutation testing was used through FSL’s Randomize tool with 5000 permutations (Winkler et al., 2014). Results were considered significant at p-family wise error (FWE) < 0.05. Results were visualized using workbench view https://github.com/Washington-University/workbench (accessed on 13 March 2017). To compare the ICs to known parcellations, each IC was parcellated into the 333 area Gordon 2016 functional parcellation for finer detail [33], and Yeo 17-resting state parcellation to assess larger network participation [32]. Non-imaging data analyses were performed in R (v. 3.6.1, Vienna, Austria) [34]. To visualize the interactions, the top and bottom HbA1c and BMI tertiles were used to graphically represent visual relationships between AnxDepr or Depr scores and the Pearson correlations between ICs.

## 3. Results

### 3.1. Demographics

Overall, the sample (n = 325) was predominately white young adults (68% white; 28.26 ± 3.91 years; Table 1).

### 3.2. Relationship between Anxiety and Depression, Depression, HbA1c, and BMI

HbA1c was negatively related to AnxDepr (d = −0.24, CI = −0.46, −0.02) and to Depr (d = −0.29, CI = −0.52, −0.07), whereas BMI was neither related to AnxDepr nor Depr (Table 2).

### 3.3. Brain Correlation Results

The associations between the interactions of HbA1c and BMI with AnxDepr and Depr on FCs between brain regions are summarized in Table 3. Overall, correlations with the cerebellum (IC 8), a visual network made up of the second visual area and dorsal stream visual area (IC 1), somatomotor mouth network (IC 13), and somatomotor hand network (IC 11) were repeatedly related to the interactions between both HbA1c and BMI with AnxDepr. Figure 1 visualizes the interaction between HbA1c and AnxDepr with FCs, including IC 8. The interaction between HbA1c and Depr was not related to any FCs.

#### 3.3.1. Unique Relationships with HbA1c and Anxiety and Depression

The interaction between HbA1c and AnxDepr was uniquely positively associated with FCs between the visual networks in ICs 1 and 4, a somatomotor network (IC 11), and cerebellar areas in IC 8 (IC 1: t = 3.76, pFWE = 0.009; IC 4: t = 3.40, pFWE = 0.035), whereas the interaction between HbA1c and AnxDepr was negatively associated with FC between the frontoparietal network (IC 7) and the cerebellar network (IC 8; t = −3.451, pFWE = 0.027).

#### 3.3.2. Unique Relationships with BMI and Anxiety and Depression

The interaction between BMI and AnxDepr was negatively associated with FCs between the DMN in IC 2 and cerebellum in IC 8 (t = −3.497, pFWE = 0.025).

#### 3.3.3. Unique Relationships with BMI and Depression

The interaction between BMI and depression was uniquely and positively associated with FC between the frontoparietal network (IC 5) and DMN (IC 14; t = 3.565, pFWE = 0.018), as well as within the frontoparietal network between ICs 5 and 7 (t = 3.37, pFWE = 0.036).

#### 3.3.4. Differences in the Interaction of BMI and HbA1c with Anxiety and Depression

AnxDepr interactions with both HbA1c and BMI, respectively, were positively associated with correlations between a visual network (IC 4) and cerebellum (IC 8), and mouth somatosensory network (IC 13) and cerebellum (IC 8). Both interactions were negatively associated with correlations between a visual network (IC 1) and DAN (IC 11), a visual network (IC 1) and mouth somatosensory network (IC 13), and DAN (IC 11) and mouth somatosensory network (IC 13). Differences in model parsimony were evaluated using AIC and are summarized Table 4.

#### 3.3.5. Similarities between Models

FCs between visual network IC 1 and somatosensory mouth IC 13, early visual network IC 4 and the cerebellum IC 8, somatosensory mouth IC 13 and the cerebellum IC 8, somatomotor and dorsal attention IC 11 and somatosensory mouth IC 13 were associated with the interactions between HbA1c and AnxDepr, BMI and AnxDepr, and BMI and Depr, respectively. Both interactions with HbA1c and BMI with AnxDepr were negatively associated with FCs the visual network IC 1 and somatosensory network and dorsal attention network IC 11. Differences in model parsimony were evaluated using AIC and are summarized in Table 4 and Table 5. Within AnxDepr, the interaction with HbA1c produced a more parsimonious model in all shared FCs except between the early visual network IC 4 and cerebellar network in IC 8 (Table 4). This suggests that despite similarities, HbA1c provides a better fit to explain the associations between ICs 13 and 8, ICs 1 and 11, ICs 1 and 13, and ICs 13 and 11. However, the interaction between BMI with AnxDepr is the best fit for examining FCs between ICs 4 and 8 when compared to the interaction of HbA1c with AnxDepr (Table 4) and BMI with Depr (Table 5).

## 4. Discussion

The present analyses sought to evaluate functional correlations (FCs) between blood oxygen-level-dependent (BOLD) response independent components (ICs) and interactions between hemoglobin A1c (HbA1c, a metric of glycemic control), body mass index (BMI), anxiety and depression, and depression in a large cohort of healthy adults from the HCP. We used four models to assess the relationship between FCs and (1) the interaction of HbA1c with anxiety and depression; (2) the interaction of BMI with anxiety and depression; (3) HbA1c with depression; and (4) BMI with depression. While our initial hypotheses were not supported, we found that the cerebellar network (IC 8) appears to be highly inter-related with cerebral networks and associated with interaction between both HbA1c and BMI with anxiety and depression. The interaction between HbA1c and anxiety and depression were uniquely and positively associated with FCs between the cerebellar IC 8 and somatosensory and dorsal attention network (IC 11)., whereas the interaction of HbA1c with anxiety and depression were uniquely negatively associated with FCs between the cerebellar IC 8 and frontoparietal network (IC 7). Furthermore, the interaction of HbA1c with anxiety and depression were negatively associated with FCs within the DMN (IC 10). The interaction between BMI and anxiety and depression was uniquely negatively related to FCs between DMN (IC 2) and cerebellum (IC 8). Finally, the interaction between BMI and depression was positively related to FCs between the frontoparietal network and DMN, as well as within the frontoparietal network. We did not see a relationship between any FCs and the interaction of HbA1c and depression. Overall, our results suggest a unique influence of anxiety and depression with glycemia on cerebellar correlations with the frontoparietal, DMN, and somatomotor networks. Given the lack of results with depression alone, this may indicate a specific relationship between anxiety and depression and glycemia in the brain.

Although the HCP sample was largely healthy with no overt diabetes or mental health diagnoses, HbA1c, BMI, anxiety and depression, and depression were inter-related. Cross-sectional linear regression revealed negative relationships between HbA1c and ASR anxiety and depression percentiles and ASR depression T-scores. While a large meta-analysis indicated a relationship between depression and type 2 diabetes (T2D) [35], studies examining HbA1c and depression show either no relationship [36,37] or a relationship with a duration of T2D [38]. Few studies have assessed the relationship between HbA1c and mental health in generally healthy samples. Darand and colleagues evaluated the relationship between dietary insulin index and depression, anxiety, and stress, and found no relationship with depression or anxiety. However, Darand did show a negative relationship between stress with dietary insulin index [39]. Although stress, anxiety, and depression are distinct, the hypothalamic–pituitary–adrenal axis and cortisol are implicated in all three [40]. Serum and plasma cortisol levels are elevated in individuals with depression compared to controls [40]. The intake of high-sugar and high-fat foods (so-called “comfort foods”) inhibits cortisol reactivity to stress [41]. Since the present sample does not have any form of diabetes, it is possible that an initial higher HbA1c, potentially from the intake of comfort foods, is associated with decreased anxiety and depression. As HbA1c approaches prediabetic or diabetic levels, the association may reverse with increased depression and anxiety, potentially due to depression and anxiety from diagnosis.

Although HbA1c was associated with lower anxiety and depression percentiles and lower depression T-scores, changes in cerebellar FC with the interaction between HbA1c and depression and anxiety, as well as the interaction between BMI and depression and anxiety, show similarities to changes seen in T2D. IC 8 was repeatedly correlated with cerebral brain regions throughout our analyses. IC 8 is made up of the cerebellum with the highest BOLD response in crus I, a superior posterior fissure, and a horizontal fissure. The cerebellum is a region with a high insulin receptor density [42]. The cerebellar insulin receptor stimulates glucose uptake and the translocation of the GLUT 4 receptor to the cell surface; additionally, the cerebellar GLUT 4 receptor is increased after prolonged exercise similar to skeletal muscle [43]. Thus, the cerebellum may be insulin-sensitive in a manner similar to skeletal muscle insulin sensitivity, and potentially relaying insulin sensitivity information elsewhere in the brain.

Increasingly, the cerebellum is recognized as more than a center of motor function [44]. Anatomical tracing has shown bidirectional projections between the cerebellum and prefrontal cortex [44]. Further, Buckner and colleagues mapped the cerebellum, showing regions corresponding to cerebral functional networks [45]. As such, we averaged IC 8′s BOLD response per parcel using Buckner and colleagues’ cerebellum atlas [45]. The highest BOLD response was within regions associated with the frontoparietal (comprised of crus I, superior posterior fissure, ansoparamedian fissure, and VIIB) and DMN (comprised of crus I and II, and horizontal fissure) [46].

The cerebellar crus I and lobule VII are implicated in depression and T2D. Presently, the interaction of HbA1c with anxiety and depression percentiles were associated with FCs between IC 8 and IC 1, IC 4, IC 7, and IC 13. IC 1 and IC 4 comprise distinct visual networks, with IC 1 encompassing the visual areas 2, 3, and 4 as well as dorsal visual stream areas (visual areas 3a), the fusiform face complex, and posterior parietal visual areas. IC 4 is made up of the early visual areas, a superior parietal cortex, and the dorsal visual stream (visual areas 7, 3b, 3cd, 4). With increasing anxiety and depression, individuals with a higher HbA1c show stronger connections between IC8 and both visual ICs 1 and 4, respectively. Previous work has shown changes between visual network and cerebellar regions in individuals with T2D [47] preceding T2D-related microvascular changes that are known to impair eyesight [47]. Functional MRI studies in individuals with diabetic retinopathy show higher regional homogeneity values in the cerebellum, a measure of connection efficiency, although in a region of interest below our findings [48]. Thus, based on previous research, the increase in connectivity between cerebellar and visual networks may indicate an early network compensation even when HbA1c is in the healthy range. In addition to the visual networks ICs 1 and 4, we saw increased FCs between IC 8 and IC 13 (made up of the mouth somatosensory area) in individuals with a higher HbA1c and anxiety and depression. Adolescents with an increased anxiety show similar patterns of increased connectivity between the cerebellum and sensorimotor network, although this was using a region of interest in the dentate nucleus distinct from where we saw a high BOLD activation in IC 8 [49]. Longitudinal studies are needed to delineate causal effects. Unlike the FCs between IC 8 and ICs 1, 4, and 13, increasing anxiety and depression and HbA1c was related to decreased FCs between the cerebellum and frontoparietal networks. The frontoparietal network, alternatively referred to as the central executive network, is involved in cognitively demanding tasks [50]. Individuals with post-traumatic stress disorder (PTSD) show a decreased cerebellar and frontoparietal network connectivity compared to controls [51]. Although PTSD and depression and anxiety are distinct, both are highly comorbid, further suggesting a shared pathway [52]. As mentioned previously, the highest BOLD response in the cerebellum was in regions associated with the frontoparietal network and DMN. The connection between the cerebellar frontoparietal network representation and the cerebral frontoparietal network has been implicated in understanding and predicting communicative actions [53]. Understanding and predicting communicative actions can be altered in individuals with depression, for example, through placing emphasis on negative predictions, the predicted negative thoughts of others (for example, “I know the other person thinks poorly of me”), and believing others’ negative behaviors are because of oneself [54]. Future research should evaluate frontoparietal and cerebellar communication among individuals with prediabetes or T2D to determine if poor glycemic control perpetuates negative thinking.

Because of high co-morbidity among HbA1c, BMI, and anxiety and depression, we compared relationships between HbA1c and BMI on anxiety and depression, and depression alone. The interaction of HbA1c with depression alone was not related to any differences in FCs. This suggests that glycemia may be more related to anxiety, with anxiety driving the associations between the interaction of HbA1c with anxiety and depression and FCs. Animal studies have shown that hypoglycemia triggers anxiety-like behaviors [55]. Most likely, the increase in anxiety with hypoglycemia promotes glucose consumption. We hypothesize that persistent anxiety is decoupled from glycemia but continues to promote sugar ingestion in an effort to soothe the anxiety.

Overall, comparisons between HbA1c and BMI interactions with anxiety and depression indicate that for all except the FCs between IC 4 and IC 8, the interaction between HbA1c and anxiety and depression resulted in a more parsimonious model. This suggests that, except for the FCs between IC 4 and IC 8, HbA1c and anxiety and depression better fit the data presented here. Alternatively, since HbA1c and depression did not interact with any FCs, depression alone may be more related to BMI rather than HbA1c. While HbA1c is a specific measure of glycemia, BMI is less specific and may be indicating other metabolic changes (change in cholesterol or inflammation). Additionally, BMI is physically overt, in that a person with a high BMI often appears physically different compared to a person with a low BMI, whereas differences in glycemia are not necessarily physically apparent. This means that BMI may also encompass body image, which has also been related to depressive symptoms [56].

The cross-sectional results from the present analysis must be interpreted within their limitations. Causal effects cannot be elucidated, thus changes in the brain may precipitate changes in mental and/or metabolic health or vice versa. The present study should be replicated in longitudinal datasets from HCP and/or the Adolescent Brain Cognition and Development study to assess temporal dynamics. Additionally, the HCP dataset, while large, is not representative of the US population. The sample is predominantly white and non-Hispanic; thus, the present results may not represent other populations. Additionally, given the self-reported nature of the height and weight measurements used to determine BMI, these results should be interpreted cautiously as individuals tend to under-report weight and overestimate height, although in young adults, self-reported height and weight can be used to calculate BMI for classification purposes [57] and the BMI data from this dataset has been published previously [58,59,60,61]. Finally, the sample studied here is largely healthy with very few individuals meeting the criteria for prediabetes. As indicated in the literature, brain FC changes with the degree and duration of high glycemia [62]. Caution should be used when interpreting the present results in terms of individuals with T2D or prediabetes. However, the present study is novel as it examined a large cohort of largely healthy individuals ameliorating any confounding from the duration of high glycemia. Further, the present analysis delineates changes in FCs from HbA1c or BMI, which is often confounded when evaluating individuals with prediabetes or T2D.

## 5. Conclusions

In summary, we found that the cerebellar regions associated with the visual network and FPN are related to the interaction of HbA1c with anxiety and depression. The interaction of HbA1c with anxiety and depression in healthy individuals may indicate that the FCs between the cerebellum and visual, somatosensory, and FPN are sensitive to mental and metabolic health. The lack of results between the interaction of HbA1c and depression alone with FC further suggests that glycemia is more related to anxiety, whereas BMI appears to interact with depression on several FCs. Further studies should further evaluate the role of the cerebellum in metabolic and mental health conditions.

## Figures and Tables

**Figure 1 brainsci-13-01086-f001:**
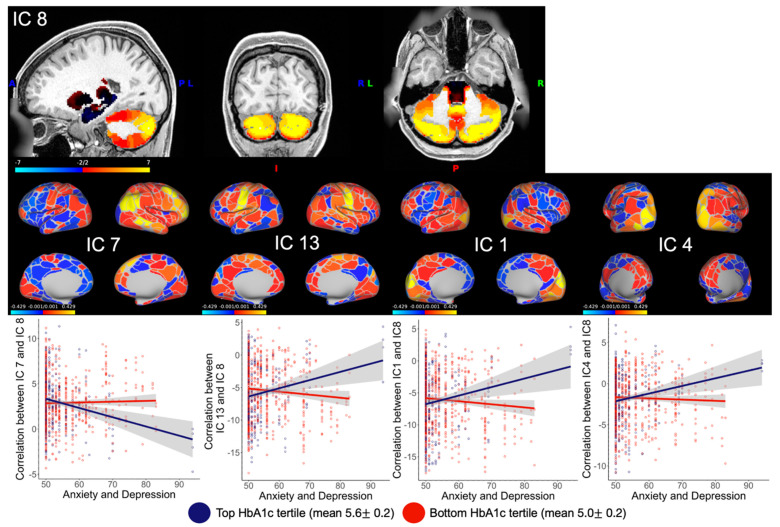
Interaction between HbA1c and Achenbach adult self-report (ASR) anxiety and depression percentile score adjusted for gender and age is associated with functional correlations between the cerebellum and frontoparietal networks, visual networks, and somatosensory mouth network. Functional ICs were parcellated using the Glasser 2013 to summarize the BOLD response into discrete parcels for ICs 7, 13, 1, and 4. IC 8 was parcellated using Buckner 200# cerebellar atlas. Graphs show the change in correlation between ASR anxiety and depression percentile scores and functional correlation between two ICs. Only the top and bottom tertiles were graphed to visualize the interaction. HbA1c was analyzed using all values on a continuous scale.

**Table 1 brainsci-13-01086-t001:** Demographics.

Metric	Mean (SD)
Age (years)	28.26 (3.91)
BMI	26.11 (4.63)
HbA1c	05.26 (0.29)
SSAGA depressive symptoms	01.48 (2.76)
ASR anxiety and depression percentile adjusted for age and gender	54.69 (7.28)
ASR DSM depression T-score adjusted for age and gender	54.60 (6.77)
Head motion	0.08 (0.03)
Gender (F/M) ^1^	144/181
*Race* ^1^	
White	221
Black or African American	48
Asian, Native Hawaiian, Other Pacific Islander	27
Native American, Native Alaskan	1
More than one	16
Unknown or not reported	12
*Ethnicity* ^1^	
Hispanic/Latinx	69
Not Hispanic/Latinx	255
Not known/Not reported	1

^1^ Frequencies.

**Table 2 brainsci-13-01086-t002:** Linear relationship between depression and anxiety and depression with HbA1c and BMI.

	Beta	SE	T	P
*ASR anxiety and depression percentile (AnxDepr)*				
HbA1c	−0.14	1.31	−2.59	0.01
BMI	0.08	0.11	1.24	0.22
SSAGA depressive symptoms	0.44	0.13	8.64	<0.0001
Age (y)	−0.02	0.09	−0.48	0.63
Gender	0.03	0.75	0.58	0.56
Black or African American	−0.03	1.08	−0.57	0.56
Asian, Native Hawaiian, Other Pacific Islander	−0.08	1.37	−1.61	0.11
Native American, Native Alaskan	0.00	6.64	0.02	0.98
More than one	0.09	1.70	1.81	0.07
Unknown or not reported	0.02	1.97	0.38	0.70
Head motion	−0.07	14.93	−1.04	0.29
	Beta	SE	T	P
*ASR DSM depression T-score (Depr)*				
HbA1c	−0.11	1.21	−2.13	0.03
BMI	0.09	0.10	1.41	0.16
SSAGA depressive symptoms	0.46	0.12	9.14	<0.0001
Age (y)	−0.05	0.09	−1.09	0.26
Gender	0.03	0.69	0.52	0.61
Black or African American	−0.03	0.10	−0.54	0.59
Asian, Native Hawaiian, Other Pacific Islander	−0.07	1.27	−1.31	0.19
Native American, Native Alaskan	0.00	6.15	0.06	0.95
More than one	0.06	1.58	1.24	0.21
Unknown or not reported	0.00	1.83	0.10	0.92
Head motion	−0.03	13.84	−0.41	0.68

**Table 3 brainsci-13-01086-t003:** Summary of brain correlation results by interaction.

IC	IC	t	pFWE
*HbA1c* × *AnxDepr*			
IC 1 Visual network (Second visual area and dorsal stream visual area)	IC 8 Cerebellum	3.756	0.009
IC 1 Visual network (Second visual area and dorsal stream visual area)	IC 11 Somatomotor/dorsal attention (Somatomotor hand)	−3.868	0.007
IC 1 Visual network (Second visual area and dorsal stream visual area)	IC 13 Somatomotor (Somatomotor mouth)	−4.562	0
IC 4 Visual network (Early visual cortex and superior parietal cortex)	IC 8 Cerebellum	3.399	0.035
IC 7 Frontoparietal (Inferior parietal cortex)	IC 8 Cerebellum	−3.451	0.027
IC 11 Somatomotor/dorsal attention (Somatomotor hand)	IC 8 Cerebellum	5.339	0
IC 13 Somatomotor (Somatomotor mouth)	IC 8 Cerebellum	4.291	0
IC 11 Somatomotor/dorsal attention (Somatomotor hand)	IC 13 Somatomotor (Somatomotor mouth)	−4.472	0.001
BMI × AnxDepr			
IC 1 Visual network (Second visual area and dorsal stream visual area)	IC 11 Somatomotor/dorsal attention (Predominantly somatomotor hand)	−3.794	0.01
IC 1 Visual network (Second visual area and dorsal stream visual area)	IC 13 Somatomotor (Predominantly somatomotor mouth)	−3.809	0.009
IC 2 Default mode network	IC 8 Cerebellum	−3.497	0.025
IC 4 Visual network (Early visual cortex and superior parietal cortex)	IC 8 Cerebellum	3.873	0.007
IC 2 Default mode network	IC 8 Cerebellum	−3.497	0.025
IC 13 Somatomotor (Somatomotor mouth)	IC 8 Cerebellum	3.683	0.014
IC 11 Somatomotor/dorsal attention (Somatomotor hand)	IC 13 Somatomotor (Somatomotor mouth)	−3.351	0.037
BMI × Depr			
IC 1 Visual network (Second visual area and dorsal stream visual area)	IC 13 Somatomotor (Somatomotor mouth)	−3.394	0.029
IC 4 Visual network (Early visual cortex and superior parietal cortex)	IC 8 Cerebellum	3.406	0.033
IC 5 Frontoparietal (Frontal–parietal and dorsal attention)	IC 7 Frontoparietal (Inferior parietal cortex)	3.37	0.036
IC 5 Frontoparietal (Frontal–parietal and dorsal attention)	IC 14 Default mode network/frontoparietal (Dorsolateral prefrontal cortex)	3.565	0.018
IC 13 Somatomotor (Somatomotor mouth	IC 8 Cerebellum	4.05	0.004
IC 13 Somatomotor (Somatomotor mouth)	IC 11 Somatomotor/dorsal attention (Somatomotor hand)	−3.646	0.012

**Table 4 brainsci-13-01086-t004:** Comparison of model parsimony between anxiety and depression’s interaction with either HbA1c or BMI.

Correlations	HbA1c AIC	BMI AIC
IC 4 and IC 8	6559.89	6556.45 *
IC 13 and IC 8	7430.92 *	7435.75
IC 1 and IC 11	6887.34 *	6887.90
IC 1 and IC 13	6622.77 *	6629.03
IC 13 and IC 11	6763.75 *	6772.48

* Indicates which AIC value is smaller and therefore more parsimonious.

**Table 5 brainsci-13-01086-t005:** Comparison of model parsimony between the interaction of BMI with either depression or anxiety and depression.

Correlations	Depression AIC	Anxiety and Depression AIC
IC 4 and IC 8	6558.98	6556.45 *
IC 13 and IC 8	7432.27 *	7435.75
IC 1 and IC 11	6891.25	6887.90 *
IC 1 and IC 13	6631.80	6629.04 *
IC 13 and IC 11	6770.57 *	6772.48

* Indicates which AIC value is smaller and therefore more parsimonious.

## Data Availability

Data are available through the Human Connectome Project https://www.humanconnectome.org/study/hcp-young-adult (accessed on 13 March 2017).

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
