# Peer review of "The Interaction of Glycemia with Anxiety and Depression Is Related to Altered Cerebellar and Cerebral Functional Correlations"

_brainsci, 2023, doi:10.3390/brainsci13071086_

Round 1
Reviewer 1 Report
Summary:
Overall, this is a very interesting manuscript and tackles some important questions about the directionality of obesity and mental health. The dataset and the functional connectivity analyses are strengths of the article, and the author has done a good job of being succinct when explaining these complex associations. However, at times it feels like the explanations are too brief. In the introduction, there is little discussion as to what the implications of brain regions in functional connectivity studies actually means. In the methods, the author has not included a description of which assessments were used to collect anxiety, depression, glycemia, or even how a simple BMI was measured. The results and discussion are well written. However it would strengthen the manuscript and the conclusions to actually run the GLM on the interaction with anxiety (without depression) to confirm the conclusions the authors are making.
Introduction:
· Line 49: missing “to”
o related “to” higher functional connectivity between the anterior
· Line 50: “Similarly, youth who experienced abuse”
o Similar to what? The previous sentence discussed youth with depression and insulin resistance and that this was related to higher functional connectivity between the ACC and the HC. However, this new sentence is discussing childhood abuse and higher fasting glucose. If these results are indeed similar they should be expanded up on to show how.
· Line 52: Meta analyses show altered connectivity in 52 the insula associated with anxiety [13], T2D [14], and obesity [15].
o Can you expand on this statement. What was the altered connectivity? Compared to controls? What was the typical connectivity? Was this a seed-to-voxel connectivity study, if so, what was the connection with the insula?
· Line 55-57: The conclusion of this paragraph is fine, but needs extra support earlier when discussing the connectivity results.
· The caudate’s function is defined, but the HC, the ACC, and the insula are not defined by their function.
· As this is the Brain Sciences journal, the author should give more background on glycemic control. It should not be assumed that the reader’s know what this is when it is mentioned on line 58.
Methods:
· For the rs-fMRI scans, the author mentions only what appears to be an eyes-open condition. Was only eyes-open used? Why not eyes-closed?
· The methods for the rs-fMRI are detailed. However, there is missing information on the other variables. How was BMI collected? How was HbA1c collected? How was anxiety and depression measured?
· Was there any other measure of obesity collected apart from BMI? DXA, Bod Pod, waist circumference, etc? This would be valuable to use in the study has included it in the data collection.
· In Models 1 and 2, why is AnxDep included twice? In models 3 and 4, Dep is included twice. Can this be explained? BMI and HbA1c is also included twice without explanation. If these GLMs are exploring the interaction terms, perhaps the inclusion on brackets would be useful
Results:
· Has the author combined anxiety and depression into one variable? This is confusing (e.g., line 141 “AnxDep”). If the author had mentioned in the methods how these variables were collected (which subset of questionaries?), then we could make an informed decision about combining these into one variable. But, this information is not provided so I am reluctant to accept the combined variable as it is, without further explanation.
Discussion:
· Line 230: “Given the lack of results with depression alone, this 230 may indicate a specific relationship between anxiety and glycemia in the brain.”
o This explanation is interesting considering that anxiety was never considered in the models alone. If anxiety was considered, then the authors may be able to claim this sentence. However, this conclusion, without analysing the anxiety data, does not sit well. I would urge the authors to include the GLM that includes the interaction with anxiety in order to confirm this analysis.
Author Response
Thank you for the in depth and helpful review. I have incorporated your comments and reviews into the manuscript. Overall, it appears that the model and specifically how Anxiety and Depression were measured was overlooked, potentially due to my own brevity. Importantly, the ASR anxiety and depression measurement measures both anxiety and depression. The anxiety and depression variable is not a combination of two metrics.
In section 2.1 we describe that the Achenbach Adult Self Report (ASR) anxiety and depression percentile score adjusted for gender and age (AnxDepr), ASR DSM depressive problems (scale) gender and age adjusted T-score (Depr), and semi-structured assessment for the genetics of alcoholism (SSAGA) depressive symptoms were used as our measures of anxiety and depression, depression, and depression symptoms respectively. To make this more overt, we have additionally added a paragraph in section 2.2 with citations to the behavioral data collection measures and the HCP protocol.
Below is a point by point response to each review.
- Line 49: missing “to”
This has been updated
- Line 50: “Similarly, youth who experienced abuse” Similar to what? The previous sentence discussed youth with depression and insulin resistance and that this was related to higher functional connectivity between the ACC and the HC. However, this new sentence is discussing childhood abuse and higher fasting glucose. If these results are indeed similar they should be expanded up on to show how.
Line 50 has been edited for clarity.
- Line 52: Meta analyses show altered connectivity in 52 the insula associated with anxiety [13], T2D [14], and obesity [15]. Can you expand on this statement. What was the altered connectivity? Compared to controls? What was the typical connectivity? Was this a seed-to-voxel connectivity study, if so, what was the connection with the insula?
We have clarified line 52 to include the direction of change in the insula and where the insula is connected to in the cited meta analyses.
- Line 55-57: The conclusion of this paragraph is fine, but needs extra support earlier when discussing the connectivity results.
We believe the updates to lines 50 though 56 increase the support earlier in the paragraph.
- The caudate’s function is defined, but the HC, the ACC, and the insula are not defined by their function.
The author assumes HC is the hippocampus, functional definitions have been added for both the hippocampus and anterior cingulate cortex.
- As this is the Brain Sciences journal, the author should give more background on glycemic control. It should not be assumed that the reader’s know what this is when it is mentioned on line 58.
We have added lines 36-41 to introduce and give additional background on glycemic control.
Methods:
- For the rs-fMRI scans, the author mentions only what appears to be an eyes-open condition. Was only eyes-open used? Why not eyes-closed?
The author was not part of the HCP and did not have say as to why eyes open or closed was used. Only eyes open was used. From my previous work as an MRI tech, eyes closed during a resting state MRI can result in people falling asleep in the scanner. The eyes open condition helps the MRI tech determine if a participant has fallen asleep.
- The methods for the rs-fMRI are detailed. However, there is missing information on the other variables. How was BMI collected? How was HbA1c collected? How was anxiety and depression measured?
The non-MRI data collection has also been published and the citation has been added along with brief summaries of how key variables were measured in the data description and preprocessing section.
- Was there any other measure of obesity collected apart from BMI? DXA, Bod Pod, waist circumference, etc? This would be valuable to use in the study has included it in the data collection.
Unfortunately, the HCP did not collect any measures of body composition.
- In Models 1 and 2, why is AnxDep included twice? In models 3 and 4, Dep is included twice. Can this be explained? BMI and HbA1c is also included twice without explanation. If these GLMs are exploring the interaction terms, perhaps the inclusion on brackets would be useful
The models are interaction models. The variables are shown twice to show that the interaction and the individual effects are both modeled. Brackets have been added to clarify the interaction terms.
Results:
- Has the author combined anxiety and depression into one variable? This is confusing (e.g., line 141 “AnxDep”). If the author had mentioned in the methods how these variables were collected (which subset of questionaries?), then we could make an informed decision about combining these into one variable. But, this information is not provided so I am reluctant to accept the combined variable as it is, without further explanation.
The AnxDepr variable is initially defined in first paragraph of the Materials and Methods in section 2.1 Sample.
“AnxDepr is not a combined variable it is the Achenbach Adult Self Report (ASR) anxiety and depression percentile score adjusted for gender and age (AnxDepr) where as the depression variable is the ASR DSM depressive problems (scale) gender and age adjusted T-score (Depr).”
We have further clarified this in section 2.2 Data description and preprocessing.
Discussion:
- Line 230: “Given the lack of results with depression alone, this 230 may indicate a specific relationship between anxiety and glycemia in the brain.” This explanation is interesting considering that anxiety was never considered in the models alone. If anxiety was considered, then the authors may be able to claim this sentence. However, this conclusion, without analysing the anxiety data, does not sit well. I would urge the authors to include the GLM that includes the interaction with anxiety in order to confirm this analysis.
Unfortunately, the HCP dataset did not measure anxiety explicitly. We have updated the sentence to read: “Given the lack of results with depression alone, this may indicate a specific relationship between anxiety and depression and glycemia in the brain.”
Reviewer 2 Report
This article was generated with data from the Human Connectome Project, WU-Minn Consortium. The research objectives were hypothesis-driven and the project has a broad level of interest due to the associations between obesity and depression and to some degree cognitive dysfunction. One focal point of the research was to investigate interactions between HbA1c and anxiety and depression revealed potential functional connections between the cerebellum and somatosensory, visual and FPN regions, suggesting their sensitivity to metabolic and mental health in healthy young people. Some results suggest chronic glycemia links to anxiety but a BMI interaction with depression. a few considerations could increase the information content of the work: 1) is there any information about the pre-enrollment diets? 2) Could waist circumferences be incorporated into the analyses? There is evidence that BMI is not the best index, and given emerging evidence about visceral obesity or fatty liver disease and cognition/behavior, waist circumference may better assess relationships with depression and anxiety; 3) The temporal lobe, hypothalamus and limbic circuits were excluded from the analyses. Please justify because those regions are critical to behavior as well. 4) The cerebellar results are interesting. Perhaps comments on the insulin sensitivity of cerebellar neurons and the age range of post-early childhood maturation, which may overlap with the age group investigated.
Author Response
Thank you for your time and review of my manuscript. I have incorporated your reviews into the present manuscript. Overall, your reviews suggest inclusion of variables that, unfortunately, the HCP group did not collect (diet, waist circumference). I agree they would strength the manuscript and hope such variables are included in future large MRI datasets. Below are point by point responses to your reviews.
- Is there any information about the pre-enrollment diets?
Unfortunately, the author is not part of the HCP consortium and did not design the assessments. The HCP data does not include any pre-enrollment diet data, although we agree this would be very informative data.
- Could waist circumferences be incorporated into the analyses? There is evidence that BMI is not the best index, and given emerging evidence about visceral obesity or fatty liver disease and cognition/behavior, waist circumference may better assess relationships with depression and anxiety;
I agree that waist circumference would be a valuable measure. Unfortunately, the HCP dataset does not have waist circumference. BMI is an adequate measure of risk of all-cause mortality particularly at the population level.
- The temporal lobe, hypothalamus and limbic circuits were excluded from the analyses. Please justify because those regions are critical to behavior as well.
Only IC 9 was excluded from the analyses to be consistent with previously published work from myself and previous lab (NUTRITIONAL NEUROSCIENCE 2021, VOL. 24, NO. 2, 140–147 https://doi.org/10.1080/1028415X.2019.1609646). Two researchers independently identified noise components from the 15 ICA netmats, with an inter-rater reliability of 93%. In the case of disagreement, the researchers reached consensus through discussion. Components were flagged as noise when BOLD activity was primarily following the gyri and/or solely following the surface of brain/skull. As a result, component 9 was determined to be noise and was removed from consideration.
- The cerebellar results are interesting. Perhaps comments on the insulin sensitivity of cerebellar neurons and the age range of post-early childhood maturation, which may overlap with the age group investigated.
I appreciate the reviewer’s suggestion and have added discussion of the cerebellar neuron insulin sensitivity to the discussion. However, due to the cross sectional nature of the analysis and the adult age range I believe discussion of post-early childhood maturation is beyond the scope of this paper, but would be interesting to investigate in future studies.
Round 2
Reviewer 1 Report
No further comments. Thank you to the author for addressing the previous issues.